



# Multiscale assessment of TRMM (3B42 V7) and GPM
# (IMERG V5) satellite precipitation products over a
# Mediterranean mountainous watershed with sparse rain
# gauges in the Moroccan High Atlas (case study of Zat basin)
Myriam Benkirane [1,*], Nour-Eddine Laftouhi [1,] Said Khabba[2], Bouabid El Mansouri[3]
1 GeoSciences Laboratory, Geology Department, Faculty of Sciences Semlalia, Cadi Ayyad University (UCAM),
Marrakech, Morocco
2 Joint International Laboratory TREMA, Physics Department, Faculty of Sciences Semlalia, Cadi Ayyad
University, (UCAM), Marrakech, Morocco
3 Natural Resources Geosciences Laboratory, Geology Department, Faculty of Sciences, Ibn Tofail University,
Kenitra, Morocco
*Correspondence to*: Myriam Benkirane (myriam14.benkirane@gmail.com)
**Abstract.** The performance of Tropical Precipitation Measurement Mission (TRMM) and its successor, Global
Precipitation Measurement (GPM), has provided hydrologists with a source of critical precipitation data for
hydrological applications in basins where ground-based observations of precipitation are sparse, or spatially
undistributed.
The very high temporal and spatial resolution satellite precipitation products have therefore become a reliable
alternative that researchers are increasingly using in various hydro-meteorological and hydro-climatological
applications.
This study aims to evaluate statistically and hydrologically the TRMM (3B42 V7) and GPM (IMERG V5) satellite
precipitations products (SPPs), at multiple temporal scales from 2010 to 2017, in a mountainous watershed under
a Mediterranean climate.
The results show that TRMM (3B42 V7) and GPM (IMERG V5) satellite precipitation products have a significant
capacity for detecting precipitation at different time steps. However, the statistical analysis of SPPs against ground
observation shows good results for both statistical metrics and contingency statistics with notable values (CC >
0.8), and representative values relatively close to 0 for the probability of detection (POD), critical success index
(CSI), and false alarm ratio (FAR). Moreover, the sorting of the events implemented on the hydrological model
was performed seasonally, at daily time steps. The calibrated episodes showed excellent results with Nash-Sutcliffe
values ranging from 53.2% to 95.5%.
Nevertheless, the (IMERG V5) product detects more efficiently precipitation events at short time steps (daily),
while (3B42 V7) has a solid ability to detect precipitation events at large time steps (monthly and yearly).
Furthermore, the modeling results illustrate that both satellite precipitation products tend to underestimate
precipitation during wet seasons and overestimate them during dry seasons, while they have a better spatial
distribution of precipitation measurements performance, which shows the importance of their use for basin
modeling and potentially for flood forecasting in Mediterranean catchment areas.
**Keywords:** Satellite precipitation, Rain gauge, Precipitation, Evaluation, Mediterranean climate, Hydrological
modeling, Zat watershed.



## 1. Introduction

Precipitation is a major force in global climate change and plays an important role in hydrological and meteorological applications (Yuan *et al.,* 2017). As a significant phenomenon in nature; precipitation has complex characteristics of spatiotemporal variations. It is one of the critical components of the global exchange of the surface material, the hydrological cycle, and disaster prevention (Bollasina *et al.,* 2011; Zhu *et al.,* 2012).

The variability of precipitation in mountainous areas directly affects local agriculture and ecological environment (Xia *et al.,* 2015; Jiang *et al.,* 2017). Moreover, the heavy precipitation events that occurred in mountainous areas frequently generate flash floods (Borga *et al.,* 2010). Therefore, the acquisition of reliable and accurate precipitation information in mountainous areas is of great significance to social and economic development and related scientific researches (Germann *et al.,* 2006). Rain gauge observation could provide a moderately accurate method for point-based precipitation measurement. However, rain gauges in mountainous regions are often scarce, irregular, and sometimes unavailable (Xia *et al.,* 2015; Hrachowitz *et al.,* 2011). Thus, in the applications that need high spatiotemporal resolution precipitation data, such as flood disaster forecasts, gauge data are regularly insufficient (Mei *et al.,* 2014; Yi *et al.,* 2018). Contrary to rain gauge precipitations, satellite remote sensing has the advantages of completely scanning the entire study region and convenient access to the data, providing an alternate way to monitor precipitation at regional and global scales (Chen *et al.,* 2018).

In recent decades, a series of high spatiotemporal resolutions Satellite Precipitation Products (SPPs), have been produced with the development of various space borne and related satellite-based precipitation retrieval algorithms, such as Artificial Neural Networks (PERSIANN) (Sorooshian *et al.,* 2000), National Oceanic and Atmospheric Administration/Climate Prediction Center (NOAA/CPC) morphing technique (CMORPH) (Joyce et al., 2004; Guo *et al.,* 2014), Climate Hazards Group InfraRed Precipitation with Station data (CHIRPS) (Funk et al., 2015), Tropical Rainfall Measuring Mission (TRMM) Multi-satellite Precipitation Analysis (TMPA) (Huffman *et al.,* 2007), and Integrated Multi-satellitE Retrievals for (GPM) mission (IMERG) (Hou *et al.,* 2014). Compared to these satellite precipitation products, the TRMM 3B42V7 precipitation product performance is higher than other products, especially in estimating extreme precipitation events in several areas around the world (Tong et al., 2014; Ringard et al., 2015). TRMM was launched in November 1997 by the National Aeronautics and Space Administration (NASA) with the collaboration of the Japanese Aerospace Exploration Agency (JXAX). The TRMM Version-7 offers quasi-global coverage (50°N–50°S) precipitation estimates at a high spatial resolution of (0.25° X 0.25°) and temporal resolution of 3 hours (Huffman *et al.,* 2007).

Given the excellent successes of the TRMM, the GPM Core Observatory satellite was set in motion by NASA and JXAX as a successor of TRMM in February 2014. Compared with TRMM, the potential of GPM to detect liquid and solid precipitation is improved by carrying space-borne dual-frequency precipitation radar (Chandrasekar *et al.,* 2015). Additionally, the GPM Core Observatory carrying a conical scanning multichannel microwave imager offers a wider measurement range (Hou *et al.,* 2014). The lately released IMERG further expands quasi-global coverage from (50°N–50°S) to (60°N–60°S) and provides precipitation estimates with a finer spatial resolution of (0.1° X 0.1°) and temporal resolution of 30 minutes (Liu *et al.,* 2017).

Since the deliverance of IMERG products, Many studies have been conducted to evaluate and compare the performance of TMPA and IMERG products regarding rain gauges observations in many regions, such USA (Gebregiorgis *et al.,* 2018), Brazil (Rozante *et al.,* 2018), Japan (Kim, et al., 2017), China(Zhong *et al.,* 2019), South Korea (Wu *et al.,* 2017), Malaysia (Tan *et al.,* 2018), Pakistan (Hussain *et al.,* 2018), South America



(Palomino-Ángel, et al., 2019), Cyprus (Retalis *et al.,* 2018), Egypt (Saber et al., 2015), and Morocco (Milewski
et al., 2015; Milewski et al.,2020). However, most of these studies indicate that IMERG had greater performance
in the characterization of precipitation variability and precipitation detection aptitude, with the only slight
improvement compared to TMPA products.
This study statistically and hydrologically evaluated GPM (IMERG V5) and TRMM (3B42 V7) satellites
precipitation estimates comparatively to ground precipitation observations over Zat semi-arid mountainous
watershed located in the Moroccan High Atlas. The objectives are to (1) Assess and statistically compare the
performance of IMERG V5 and 3B42 V7 precipitation products at multiple temporal scales in the Zat basin, (2)
Analyze the precipitation detection ability of 3B42 V7 and IMERG V5 satellite sensors and (3) Evaluate the ability
of the SPPs to reproduce rainfall events and demonstrate their aptitude to provide meaningful information in
hydrological modelling and flood forecasting.
This manuscript provides a valuable reference for monitoring and forecasting precipitation in mountainous regions
characterized by a Mediterranean climate, as well as basins where rainfall stations are scarce or poorly distributed.
**2. Study Area and Datasets**
**2.1. Study area**
Zat watershed is a sub-basin of the Tensift catchment, it's also an Atlas tributary located on the left bank of Tensift
river and situated in the Moroccan High Atlas Mountains (Mount Toubkal, the highest mountain in North Africa),
in the South EST of Marrakech city. Geographically the sub-basin is found between latitude 31°30′and 31°45′
North and longitude 7°30′ and 7°45′ West. It's drained by the Zat River, which measures 89 km, the slopes are
often very steep with an average of 19%, and it covers a total area of about 519 km² (Figure 1). The topography of
the catchment area varies from 3777 m (above sea level) downstream to the Taferiat station where the outlet is at
an altitude of 756 m. (Benkirane *et al.,* 2020).
This sub-basin is characterized by Mediterranean climate strongly influenced by altitude. Taferiat hydrometric
station controls the discharge of the Zat Basin, and also serves as rain gauge. It receives an annual rainfall average
ranges from 133 mm /year to 913 mm /year; precipitation is mainly concentrated during the rainy period from
October to April and a hot and dry period from May to September. Therefore, this study region is subject to
frequent flash floods and droughts.

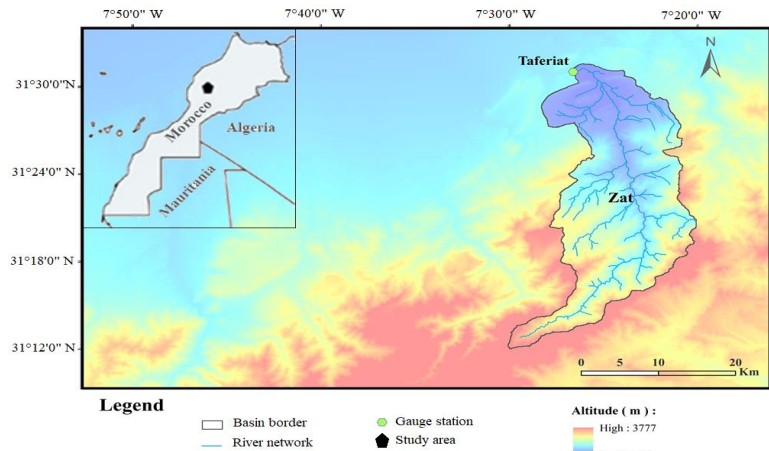




108        **Figure 1. The geographical location of the Zat basin and rain gauge station used in the study.**

**2.2. Rain gauge data**
Rain gauge measurements are daily precipitation data collected from only one meteorological station shown in
(Figure. 1) located at the outlet of Zat basin, covering a period from 2010 to 2017. Data sets are provided by the
Tensift Hydraulic Basin Agency. These data were used as a benchmark for evaluating TRMM (3B42 V7) and
GPM (IMERG V5) SPPs. All observations provided by these stations are subject to strict quality control such as
climate limit value inspection, and station extreme value inspection (Shen *et al.,* 2018). In addition, the monthly
and yearly precipitation values are accumulated from daily observations.
**2.3. Satellite precipitation data**
This study evaluated two types of satellite precipitation products (SPPs), the TRMM (3B42 V7), and the GPM
(IMERG V5) at different time scales, from September 01 2010 to August 31 2017 (Table1). Before accumulating
the dataset from daily to monthly and yearly precipitation, both products were converted from UTC to UTC +1 to
unify the time with the study area (Wang *et al.,* 2019), a brief description of these SPPs is given as follows.
**Table 1. Main parameters of TRMM (3B42 V7) and GPM (IMERG V5) satellite precipitation products.**

| Satellite precipitation products | TRMM (3B42 V7) | GPM (IMERG V5) |
|---|---|---|
| Temporal / Spatial Resolutions | 3 h /0.25° | 0.5 h /0.1° |
| Coverage Period | December 1997 to Present | March 2014 to Present |
| Coverage Range | Global (50°N–50°S) | Global (60°N–60°S) |

**2.3.1. TRMM (3B42V7)**
The TRMM (3B42 V7) precipitation products were generated by using the TRMM 3B42 Version 7 algorithm
(Huffman et al., 2007). It was designed to combine various microwaves MW, and infrared IR satellite-based
precipitation estimates with gauge adjustments observations to provide 3-hourly quasi-global quantitative
precipitation estimates *(Hou et al., 2014)*. The 3B42 V7 product is derived by bias-adjusting the near-real-time
product with the GPCC monthly gauge-analysis precipitation data set, and it has two-month latency (Yuan *et al.,*
2017). The product can produce rational precipitation estimates in a 0.25° spatial resolution with a quasi-global
coverage (50°S–50°N). In this study, the TRMM 3B42 V7 daily precipitation product was acquired from the
Precipitation Measurement Mission (PMM) website (https://pmm.nasa.gov/data-access/downloads/trmm).
**2.3.2. GPM (IMERG V5)**
The GPM project is the result of collaboration between (NASA) and (JAXA). GPM satellite carries two primary
sensors: The multi-channel GPM Microwave Imager (GMI), and the Dual-frequency Precipitation Radar (DPR).
This satellite product is expected to provide the next-generation global observations of rain and snow and to
improve weather and precipitation forecasts through the assimilation of instantaneous precipitation information
(Kim *et al.,* 2017). IMERG is the Level 3 precipitation estimation algorithm of GPM, it provides three different
daily IMERG products, which include IMERG Day 1 Early Run (near real-time with a latency of 6 h), IMERG
Day 1 Late Run (reprocessed near real-time with a latency of 18 h) and IMERG Day 1 Final Run (gauged-adjusted
with a latency of four months) products (Chen *et al.,* 2018). In this study, we evaluate the latest released GPM
IMERG Version 5 (IMERG V5), the dataset is produced at NASA Goddard Earth Sciences (GES). The IMERG
precipitation products have a relatively finer spatial 0.1°spatial resolution with spatial coverage from 60°S to 60°N
and temporal (half-hourly) resolution.  The daily precipitation data were accumulated to obtain monthly and annual





precipitation. The GPM (IMERG V5) precipitation data were downloaded from the PMM website
(https://pmm.nasa.gov/data-access/downloads/trmm).
**2.4. Methodologies**
Different methods were used to compare the IMERG V5 and 3B42 V7 products with the gauge precipitation data
from the Taferiat station, depending on the time steps considered (daily, monthly, and annual). However, the
satellite products represent the rainfall estimates at the scale of (0.1 ° for IMERG V5 and 0.25 ° for 3B42 V7,
respectively), while the gauge precipitation observed represents precipitation on a point scale. For comparison, the
method frequently used is to increase the point precipitation data from the gauges to the same grid scale as the
SPPs, either by spatial interpolation or simply by calculating the average. In addition, the researchers pointed out
that the interpolation can lead to uncertainties due to systematic error and the density of the gauge (Duan *et al.,*
2016). Therefore, a direct comparison is used in this study. To evaluate these two SPPs, we only considered the
grids that cover the data of the single gauging station present in the Zat basin. Therefore, the grids not covering
the gauge station were excluded from the assessment.
Furthermore, to evaluate the ability of the SPPs to reproduce rainfall events, it was decided to use them as input
data in a surface hydrological model, the HEC-HMS model. Indeed, the rainfall measurement stations are not
precise and poorly distributed spatially, especially in the mountainous regions of the High Atlas, which is a real
issue for research work on hydrological modeling and flood forecasting. Consequently, it is important to evaluate
the rainfall estimated by the satellites to demonstrate their ability to provide significant information and to approve
their use as an alternative source of rainfall measurement data
**2.4.1. Statistical Evaluation of Satellite Precipitation Products**
Several diagnostic indices were used to statistically assess the quality of IMERG V5 and 3B42 V7 products
compared to observations of ground precipitation. Indeed, the comparison was carried out based on a general
evaluation (continuous statistical measurements) and of the precipitation detection capacity (categorical statistical
measurements) (Table 2).
**Continuous statistical indices**
Four statistical measures were selected, including correlation coefficient (CC), root mean square error (RMSE),
relative bias (RB), and bias (bias), which were calculated to statistically evaluate the two PPS products.
The Pearson Correlation Coefficient (CC) measures the agreement between the PPS products and the gauge data.
The (RMSE) was used to represent the mean magnitude of the error. The (RB) and (bias) was applied to evaluate
the systematic bias between the SPPs and gauge data in percent and amount of precipitation, respectively. The
overestimation of the precipitation estimate is represented by positive (RB) or (Bias) values, and vice versa.
**Table 2. Statistical metrics for evaluating IMERG V5 and 3B42 V7 products**

| Statistical Index | Units | Equation |
|---|---|---|
| **Correlation Coefficient ( R )** | Ratio | $CC = \dfrac{\sum_{i=1}^{N}(Pi - \bar{P})(Si - \bar{S})}{\sqrt{\sum_{i=1}^{N}(Pi - \bar{P})^2 \sum_{i=1}^{n}(Si - \bar{S})^2}}$ |
| **Root Mean Square Error (RMSE)** | mm | $RMSE = \sqrt{\dfrac{\sum_{i=1}^{N}(Pi - Si)^2}{N}}$ |





| | | |
|---|---|---|
| **Mean Absolute Error (MAE)** | mm | $MAE = \dfrac{\sum_{i=1}^{N}|Pi - Si|}{N}$ |
| **Relative Bias (RB)** | % | $RB = \dfrac{\sum_{i=1}^{N}(Pi - Si)}{\sum_{i=1}^{N} Si} X\ 100\%$ |
| **Bias** | N/A | $Bias = \dfrac{\sum_{i=1}^{N}(Pi - Si)}{\sum_{i=1}^{N} N}$ |
| **Probability Of Detection (POD)** | Ratio | $POD = \dfrac{a}{a + c}$ |
| **False Alarm Ratio (FAR)** | Ratio | $FAR = \dfrac{b}{a + b}$ |
| **Critical Success Index (CSI)** | Ratio | $CSI = \dfrac{a}{a + b + c}$ |
| **Frequency Bias Index (FBI)** | Ratio | $FBI = \dfrac{a + b}{a + c}$ |


Where N represents the number of samples; Si and $\bar{S}$ are gauge observations and their average; Pi and $\bar{P}$ represent
satellite estimates and their average, respectively.
Also, a, denotes the number of rainfall events that observed and detected; c, is the number of rainfall that failed to
be detected by the satellite; b, denotes the number of rainfall events detected by the satellite that did not occur; the
threshold for identifying a precipitation event is 0.5 mm/day.
**Categorical statistical indices**
To evaluate the precipitation detection capability of IMERG V5 and 3B42 V7 products, four categorical statistical
indices were calculated to assess the ability of PPSs. The most common measures, counting Probability of
Detection (POD), False Alarm Rate (FAR), Critical Success Index (CSI), and Frequency Bias Index (FBI) are used
in this study. The values of all categorical statistical measures are between 0 and 1.
The POD indicated the ratio of the number of precipitation events correctly detected by satellites among all real
precipitation events. The FAR is the ratio of false alarming precipitation events to the total number of detected
precipitation events. The FBI represents the fraction of falsely detected precipitation events (false alarm) compared
to the total number of detected precipitation events, it indicates whether the dataset tends to overestimate (FBI> 1)
or underestimate (FBI <1) precipitation events. The CSI reported the number of correct predictions of a rain event
divided by the total number of successes, false alarms, and failures. Table 3 shows the formulas for these metrics.
**Table 3. Contingency table to evaluate precipitation occurrence by satellite products**

| | | Satellite | |
|---|---|---|---|
| | | Rain ( daily rain ≥0.5 mm) | No rain (daily rain <0.5 mm) |
| **Gauge** | **Rain ( daily rain ≥0.5 mm)** | a: hits | b: false |
| | **No rain (daily rain <0.5 mm)** | c: misses | d: correct negatives |





**Hydrological Model**
The Hydrologic Engineering Centre's Hydrologic Modeling System (HEC-HMS) is designed to simulate the
rainfall-runoff processes of dendritic watershed systems. This model is known to be applicable in a wide range of
geographic areas for solving the broadest possible range of problems (Scharffenberg and Fleming 2016). The
method used in this paper includes SCS-CN (Soil Conservation Service) Curve Number, Clark Unit Hydrograph,
and Baseflow Recession, which are necessary to determine the hydrologic loss rate, runoff transformation, and
base flow rates. This method aims to calibrate four rainfall events according to seasons (autumn, winter, spring,
summer), with a daily time step precipitation by implementing the model with different precipitation data sources
such as observed and satellite precipitation with observed runoff to evaluate the ability of the SPPs to reproduce
rainfall events according to seasons (Figure. 2).

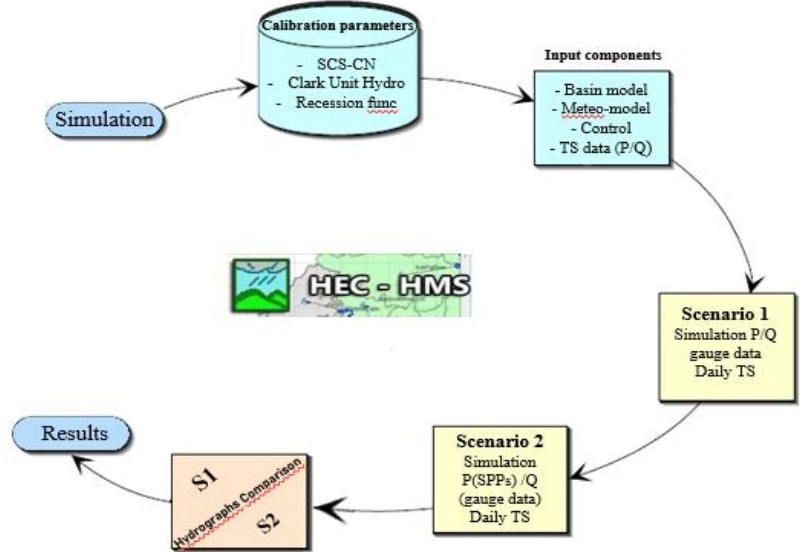


**Figure 2. Schematic representing the adopted approach for the hydrological modelling research.**
**3. Results**
**3.1. Assessment of precipitation at a different time scale**
The rainfall time series of the two selected satellite products and the rain gauge at different timescales in the Zat
basin are presented in (Figure 3). In general, the 3B42 V7 and IMERG V5 products present similar chronological
precipitation patterns to those of the gauge. However, it can be seen that the product 3B42 V7 slightly
overestimated the daily precipitation, while the product IMERG V5 showed good performance on the daily
timescale (Figure 3A). Regarding the monthly precipitation series, the product 3B42 V7 underestimated the
monthly precipitation, while IMERG V5 clearly showed good initial agreement with the observed precipitation,
although from 2015 IMERG V5 slightly overestimated the monthly precipitation (Figure 3B). As for the annual
time scale, the precipitation series was overestimated by the precipitation products 3B42 V7 and IMERG V5, this
phenomenon is similar to that of the monthly scale, the overestimation was observed from 2015 onward (Figure
3C).

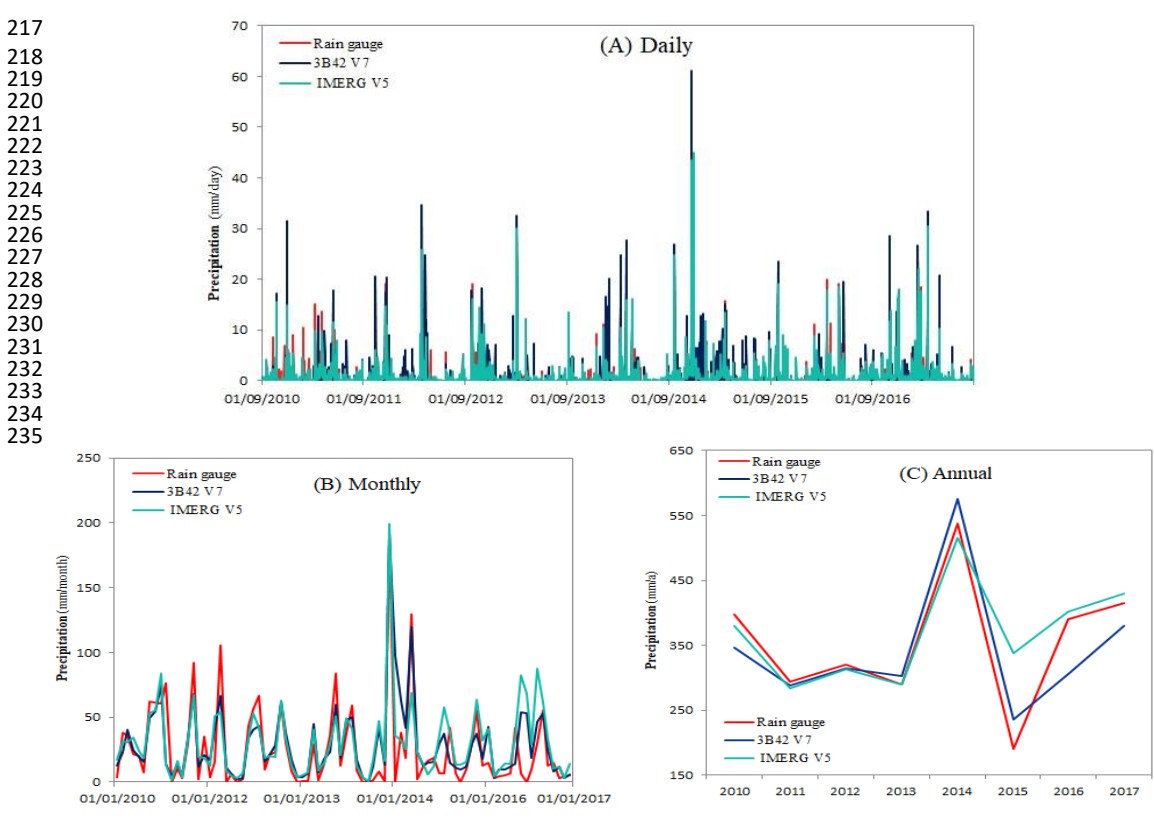

**Figure 3. Precipitation time series from rain gauges, 3B42 V7, and IMERG V5, in the Zat basin from 2010 to 2017 (a, Daily; b, Monthly; c, Annual).**

### 3.2. Statistical evaluation

The SPPs were statically compared against the ground observations to evaluate their accuracy and reliability. Table 4. Lists the evaluation results of statistical metrics (CC, RMSE, MAE, R Bias, and Bias), thought the entire study period over the Zat basin.

**Table 4. Statistical metrics results of 3B42 V7 and IMERG V5 precipitation estimates at multiple time scales from 2010 to 2017.**

|  | TRMM | | | GPM | | |
|---|---|---|---|---|---|---|
|  | **Daily** | **Monthly** | **Yearly** | **Daily** | **Monthly** | **Yearly** |
| **CC** | 0.78 | 0.80 | 0.90 | 0.81 | 0.75 | 0.85 |
| **RMSE** | 2.03 | 19.90 | 42.84 | 1.68 | 22.09 | 53.64 |
| **MAE** | 0.71 | 12.85 | 34.72 | 0.62 | 14.49 | 29.03 |
| **R Bias** | 29.48 | 16.47 | - 2.99 | 47.13 | 21.73 | 4.07 |
| **Bias** | 0.19 | 4.14 | - 10.61 | 0.31 | 5.46 | 14.44 |

Figures 4 and 5. Shows the scatterplots and boxplots with the statistical metrics for the 3B42 V7 and IMERG V5 products versus ground-based rain gauge observation. The scatterplots of SPPs products against rain gauge precipitation exhibit a concentration of the points near the 1:1 line especially at daily and monthly scales. According to the metrics plotted in Figures 4 and 5 (A), 4 and 5 (B), and 4 and 5 (C), the performance of the IMERG V5 is superior to that of the 3B42 V7 at daily scale. However, the obtained results at monthly and yearly scales for the 3B42 V7 are significantly better than IMERG V5.







**Figure 4. Scatterplots forTRMM3B42 v7 and GPM IMERG v5 precipitation versus ground-based rain gauge**
**measurements over the Zat watershed "(A–C) daily, monthly, and annual scales, the black and red oblique lines**
**denote a 1 :1 line and a least-squares regression line, respectively".**



Both 3B42 V7 and IMERG V5 products present a strong correlation with gauge data at a daily scale. Figure 4 A
shows a high CC (0.78) and (0.81) respectively a small RMSE error values (2.03 mm ) and (1.68mm) respectively,
and relatively balanced R Bias and Bias (29.4%), (0.19) for 3B42 V7 and (47.13%), (0.31) for IMERG V5. In
general, except for the R Bias and the Bias values, the other continuous statistical indices from both products had
good results, and it can be seen that IMERG V5 indices were better than 3B42 V7 at the daily scale Figures (3 and
4 A). Figures (4 and 5 B), Represents scatterplots and boxplot of precipitation from 3B42 V7 and IMERG V5 at
monthly scale. Compared with gauge data, it can be seen that both products slightly underestimate the precipitation.
However, 3B42V7 show a much better correlation with gauge precipitation than IMERG, with higher CC ( 0.80)
and (0.75), low RMSE (19.90 mm vs. 22.09mm), acceptable MAE (12.85) and (14.42), and relatively low values
of R Bias and Bias (16.47% vs. 21.73%) and ( 4.14 vs. 5.46) respectively. Meanwhile, 3B42 V7 performed better
than IMERG V5 on a monthly scale. According to the scatterplots and boxplot illustrated in Figures (3 and 4 C),
it can be noticed that the 3B42 V7 product is obviously manifested by a slight underestimation and that the major
performance of the IMERG V5  product is moderately superior to that of 3B42 V7, except at the value of CC and
RMSE. The illustration of both 3B42 V7 and IMERG V5 present a strong correlation with high CC values of
(0.90) and (0.85), low RMSE error (42.84 mm ) and ( 53.64mm), small MAE (34.72) and (29.03), and relatively
good R Bias and Bias (-2.99% vs. 4.07%) and (-10.61 and 14.44) respectively. For the R Bias and Bias, the negative
deviation of 3B42 V7 precipitation estimates was relatively balanced, while IMERG V5 showed a positive
deviation at the rain gauge. Indeed, IMERG V5 showed better performance than 3B42 V7 at a yearly time scale.

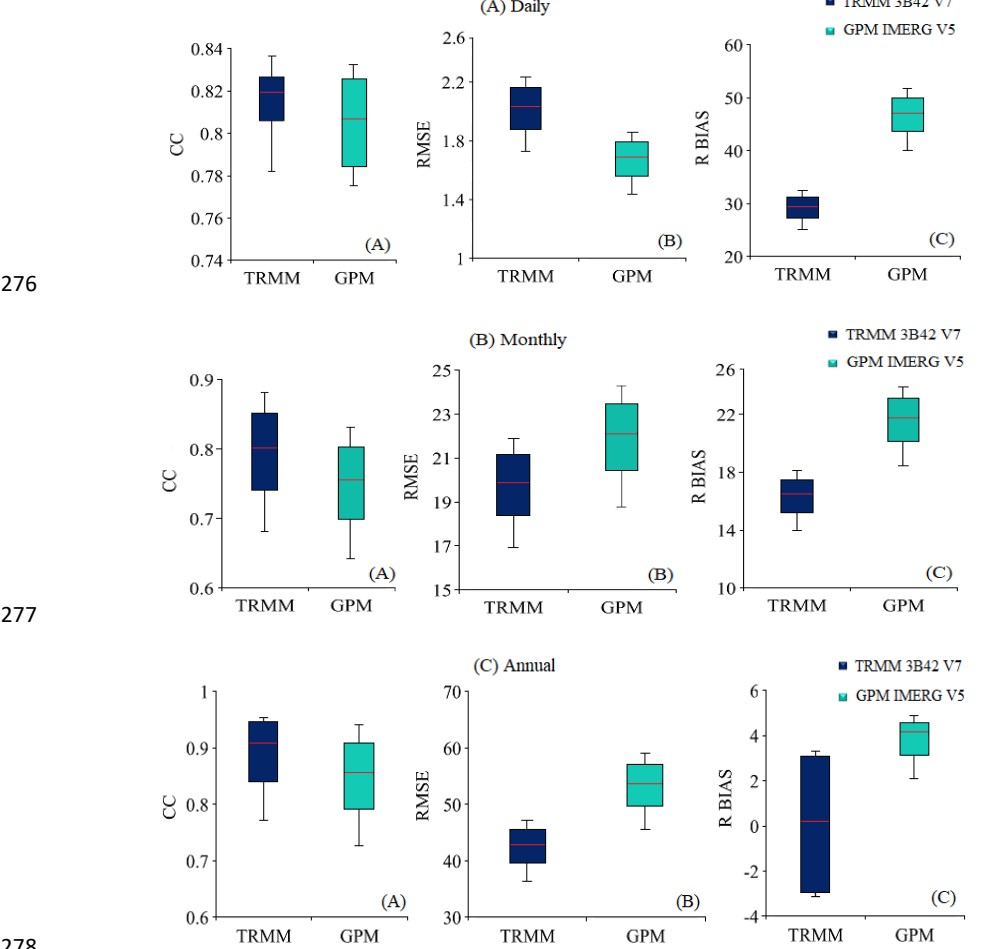







**Figure 5. Boxplot of correlation coefficient (CC), root mean square error (RMSE), and relative bias (R BIAS) between**
**satellite-based and rain gauge at multiple time scales in Taferiat gauge station, during the period of September 1st to**
**August 31, 2010–2017.**
**3.3. Contingency Statistics**
The categorical statistical metrics of 3B42 V7 and IMERG V5 at different time scales are shown in (Table 5).
**Table 5. Contingency statistical metrics results of 3B42 V7 and IMERG V5 precipitation estimates at multiple time**
**scales from 2010 to 2017.**

|  | TRMM | | | GPM | | |
|---|---|---|---|---|---|---|
|  | **Daily** | **Monthly** | **Yearly** | **Daily** | **Monthly** | **Yearly** |
| **POD** | 0.6 | 1 | 1 | 0.89 | 1 | 1 |
| **FAR** | 0.59 | 0.1 | 0 | 0.68 | 0.12 | 0 |
| **CSI** | 0.32 | 0.89 | 1 | 0.33 | 0.88 | 1 |
| **FBI** | 1.47 | 1.1 | 1 | 2.85 | 1.13 | 1 |

The precision of 3B42 V7 and IMERG V5 at daily, monthly and annual scales was compared and analyzed.
IMERG V5 demonstrated better performance than 3B42 V7 in detecting precipitation events on a daily scale, with
low values of POD and CSI (0.89 vs. 0.6) and (0.33 vs. 0.32) (Figure 5 A, B), as well as reasonably high values
of FAR and FBI (0.68 vs. 0.59) and (2.85 vs. 1.47) respectively (Figure 5 C, D).
The performance of the categorical statistical measures at the monthly level is shown in Figure 5. 3B42 V7, and
IMERG V5, produced good results for rainfall estimation, with POD values and CSI approximately similar to the
perfect values in Table 2, the respective values are ( 1 vs. 1) and (0.89 vs. 0.88) (Figure 5 A, B). Similarly for the
FAR and FBI the results obtained are close to the perfect values, (0.1 vs. 0.12) and (1.1 vs. 1.13) respectively
(Figure 5 C, D).
Regarding annual performance, IMERG V5 and 3B42 V7 products show very good results, the performance of
the POD is consistent with that of the CSI, which exhibits perfect values (1 vs. 1) and (1 vs. 1) (Figure 6 A, B) and
similarly, for the FAR and FBI (0 vs. 0) and (1 vs. 1) respectively (Figure 5 C, D), the values are perfectly adequate
to those of Table 2.

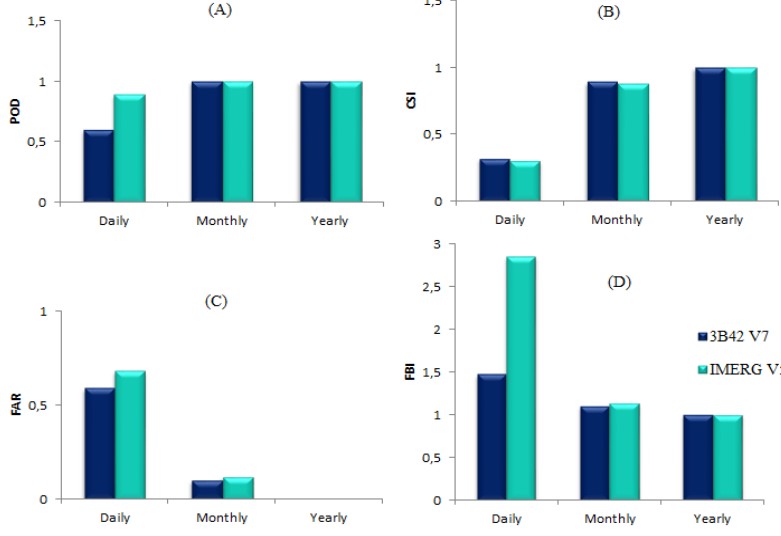


**Figure 6. Metrics results of different time scales (A) POD, (B) CSI, (C) FAR, and (D) FBI, of 3B42 V7 and IMERG V5**
**products.**





In general, IMERG V5 is better at detecting precipitation events, in particular at capturing precipitation traces and
solid precipitation at a daily scale, while 3B42 V7 can estimate precipitation on a large time scale.

## 4. Hydrological evaluation of discharge simulation using two SPPs.

The HEC-HMS model was used to calibrate the daily rainfall events from (1/09/2010) to (31/08/2017) according
to the different seasons, at the level of the Zat basin, using the rainfall and Runoff data from the Taferiat gauge
station, and satellite precipitation products, the four episodes that we chose to present are the three most
representative of the data series. The hydrological simulations were carried out according to two different
scenarios:
Scenario 1: Simulation and calibration by implementing the model with observed rainfall and discharge data.
Scenario 2: Simulation and calibration using rainfall from both satellite products with observed flows.

### 4.1 Event of November 22nd, 2014 (autumn)

This event represents a torrential flood; since the flood was generated by extreme precipitation spread over more
than 15 days, it is the most intense event in the data set. The maximum flow reached is (123,75 m3/s). However,
the soils were saturated, resulting in high permeability and an increase in the runoff coefficient of the watershed.
The results of the simulation and calibration of the rainfall data and the observed flow in the hydrograph of scenario
1 (Figure 6), show that the simulated flow curve was well reproduced both at the flood rise and the recession part,
although the peak flow was not reached, the evaluation criteria are very satisfactory: RMSE = 0.5 and Nash =
75.60%. Scenario 2 represents the results of the simulations and event calibrations with the implementation of the
model with the previously analyzed satellite precipitation products (IMERG V5, 3B42 V7) and the observed flows.
Hydrographs were well reproduced for both the rise and the recession; peak flows were achieved and evaluation
criteria are very satisfactory with RMSE of 0.5 and 0.4 and Nash of 77% and 81.11% respectively.
The simulation results of scenario 2 are better than those of scenario 1. This is explained by the fact that the
satellites have several measuring stations well distributed spatially while the Zat basin has only one measuring
station downstream (Taferiat station). In addition, a significant underestimation was found during the analysis of
the 2 SPP data compared to the observed data, this underestimation was compensated by using the elevation of the
curve number during the calibration of the event.

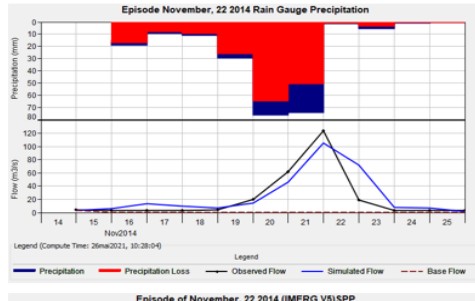

Table 6. Performance of the event of November, 22 2014 under the two scenarios

| Scenarios | Precipitation products | Curve Number | Root Mean Square Error (mm) | Nash-Sutcliffe (%) |
|---|---|---|---|---|
| I | Gauge Precipitation | 30 | 0,5 | 75,60% |
| II | TRMM (3B42 V7) | 77 | 0,4 | 81,80% |
| II | GPM (IMERG V5) | 88 | 0,5 | 77% |

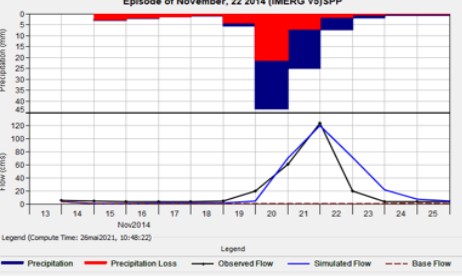

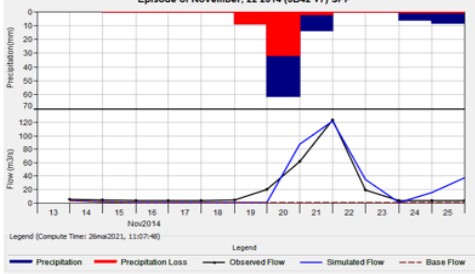

**Figure 7. Simulation of the episode of November, 22 2014, using rainfall and runoff gauge data as input (Scenario I),**
**and SPPs with measured flow data as input (Scenario II).**





### 4.2 Event of the February 28, 2016 (winter)

The event represents a winter rain storm characterized by liquid precipitation downstream and snow upstream of the watershed. This type of rain storm is very frequent during the winter, especially in the high mountains of the Atlas.

The hydrograph of scenario 1 represents a simulated flow curve quite illustrative, the rising curve and the recession were well reproduced, contrary to the peak flow which has not been reached, the evaluation criteria are moderately good representing values of RMSE = 0.6 and Nash = 63.1%, this is induced by the fact that the snowy fraction has not been taken into account due to the irregularity of the precipitation measuring stations.

On the other hand, the hyetogram of scenario 2 illustrates a good spatial distribution of precipitation, although the curves of the simulated flows are quite well reproduced, and the peak flows were not reached, the evaluation criteria are acceptable with RMSE of 0.7 for IMERG V5 and 3B42 V7 and Nash of 57.8% and 53.2% respectively. Given this, an intense underestimation of the precipitation was noticed while obtaining the calibration results, which are mainly related to the fraction of snow precipitation not considered.

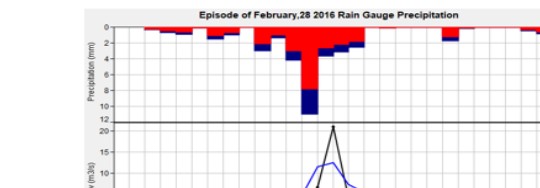

Table 7. Performance of the event of February, 28 2016 under the two scenarios

| Scenarios | Precipitation products | Curve Number | Root Mean Square Error (mm) | Nash-Sutcliffe (%) |
|---|---|---|---|---|
| I | Gauge Precipitation | 70 | 0,6 | 63,1% |
| II | TRMM (3B42 V7) | 77 | 0,7 | 53,2% |
| II | GPM (IMERG V5) | 89 | 0,7 | 57,8% |

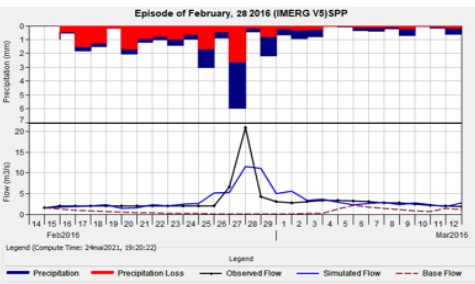
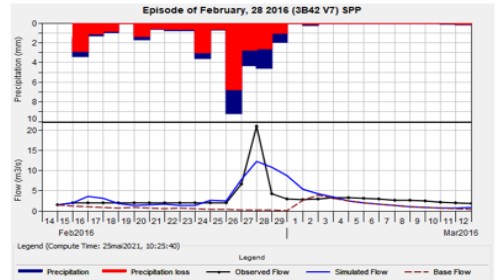

**Figure 8. Simulation of the episode of February, 28 2016, using rainfall and runoff gauge data as input (Scenario I), and SPPs with measured flow data as input (Scenario II).**

### 4.3 Event of May 02, 2011 (spring)

This event represents the typical characteristics of a freshet caused by the melting of snowfall upstream of the Zat watershed during the winter, with the progressive increase of temperatures during the spring the snow cover at the summit of the Atlas Mountains start melting and feed the streams of the mountainous basins including the study basin. This usually causes significant flooding during the occurrence of moderate rainfall episodes.  The hydrograph of scenario 1 is perfectly calibrated, the simulated flow curve was well reproduced at the rise and at the recession, the peak flow was reached, and the evaluation criteria are excellent with an RMSE of 0.2 and a Nash of 95.5%.  On the other hand, the hydrographs of scenario 2 are perfectly reproduced at both rising and recession parts, the peak flow has been reached, the evaluation criteria are excellent with an RMSE of 0.2 for the product (IMERG V5) and 0.3 for (3B42 V7), and a Nash of 87% and 73% respectively. Indeed, the overestimation of precipitation generated by the satellite products was compensated by the flows produced by the snowmelt, which allowed the model to reproduce this event well. It is essential to mention that the SPP overestimates precipitation during warm seasons.





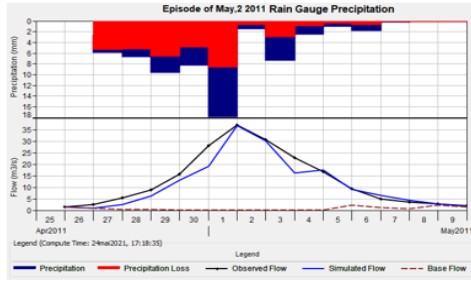

Table 8. Performance of the event of May, 02 2011 under the two scenarios

| Scenarios | Precipitation products | Curve Number | Root Mean Square Error (mm) | Nash-Sutcliffe (%) |
|-----------|------------------------|--------------|------------------------------|---------------------|
| I | Gauge Precipitation | 73 | 0,2 | 95,5% |
| II | TRMM (3B42 V7) | 83 | 0,3 | 92% |
| II | GPM (IMERG V5) | 87 | 0,2 | 94,20% |

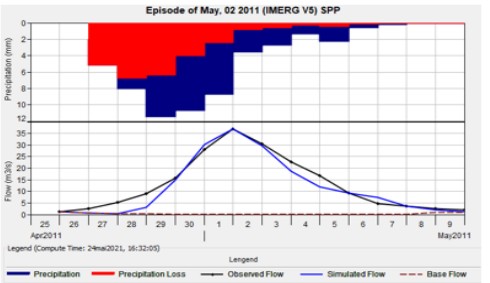

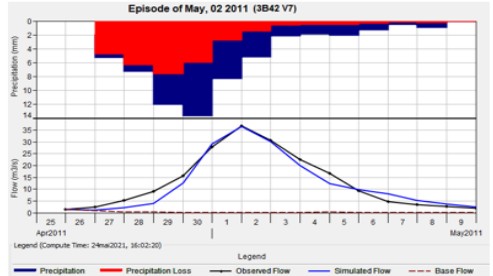

**Figure 9. Simulation of the episode of May, 02 2011, using rainfall and runoff gauge data as input (Scenario I), and SPPs with measured flow data as input (Scenario II).**

### 4.4 August 27th, 2011 event (summer)

The summer event of August 27th, 2011, called flash flood, is characterized by a sudden occurrence and short duration due to stormy precipitation, and initial soil conditions very favourable to runoff. The Atlas watersheds are mostly dry during the summer due to the high temperatures typical of the Mediterranean climate, and as such low rainfall can cause dangerous floods.

The simulated flow curve of the hydrograph of scenario 1 is well reproduced despite the slight shift located at the rise and the recession curve, as well as the peak flow that is not reached, the evaluation criteria are satisfactory, an RMSE of 0.4 and a Nash of 84.30%.

The hydrographs of scenario 2 are well reproduced, in particular, that of the SPP (3B42 V7), the evaluation criteria are satisfactory with an RMSE of 0.4 for (IMERG V5) and 0.3 for (3B42 V7), as well as a Nash of 80.1% and 91.3% respectively.

It is important to note that the basin response is relatively slow due to the initial soil conditions, additionally to the overestimation of precipitation from the satellite products during the high-temperature seasons.





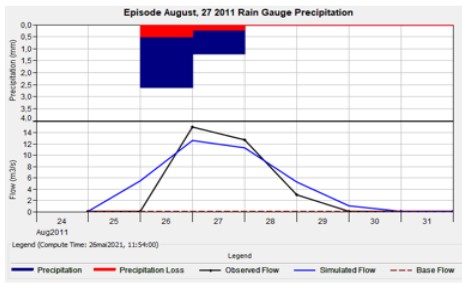

Table 9. Performance of the event of August, 27 2011 under the two scenarios

| Scenarios | Precipitation products | Curve Number | Root Mean Square Error (mm) | Nash-Sutcliffe (%) |
|---|---|---|---|---|
| I | Gauge Precipitation | 76 | 0,4 | 84,30% |
| II | TRMM (3B42 V7) | 77 | 0,3 | 91,3% |
| II | GPM (IMERG V5) | 70 | 0,4 | 80,10% |

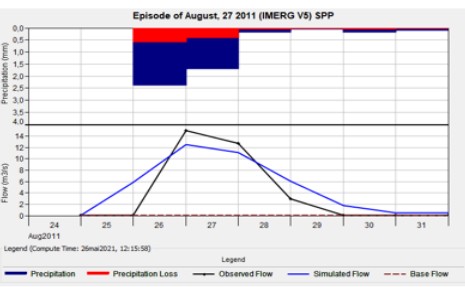

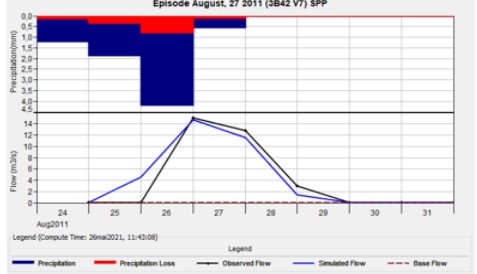

**Figure 10. Simulation of the episode of August, 27 2011, using rainfall and runoff gauge data as input (Scenario I), and SPPs with measured flow data as input (Scenario II).**

This is an efficient method developed in this paper for the first time in a country with a Mediterranean climate, on the mountainous watersheds of the Moroccan High Atlas with low density and irregularity of precipitation and flow measurement stations. This is a good method to apply to solve the problem of deficiency of observed data in these regions.

## 5. Conclusion

SPPs are important precipitation data alternatives, particularly in high mountain watersheds, where measurement gauge stations are poorly distributed or absent. These products will mainly help in the simulation of river flows, flood forecasting, and water resources management in arid to semi-arid regions. This study conducted a complete performance evaluation of two satellite products the TRMM (3B42 V7) and the GPM (IMERG V5) using observations (daily, monthly and annual) collected the only gauge station of the Zat basin, named Taferiat station and located at the downstream of the watershed. The watershed is characterized by a Mediterranean climate and mountainous topography, and the study was analyzed over 7 years, from September 01, 2010 to August 3, 2017. To evaluate the accuracy of 3B42 V7 and IMERG V5 satellite precipitation products, several quantitative, categorical, and graphical statistical measurements were used, (R, RMSE, MAE, R Bias, Bias) are used to quantitatively analyze the accuracy of satellite precipitation products, and (POD, CSI, FAR, and FBI) were used to evaluate the precipitation detection capability of satellite precipitation products, and to simulate satisfactorily the flooding events in hydrological model.
The conclusions resulting from this study are summarized as follows:
(1) IMERG V5 and 3B42 V7 products performed well in estimating daily, monthly, and annual precipitation compared to observed data from the Taferiat station. SPPs products slightly underestimated the daily and annual precipitation, while 3B42 V7 slightly underestimated the monthly precipitation, especially during winter periods.
(2) Compared to the ground applications, 3B42V7 and IMERG V5 showed good correlation results at the daily scale. However, IMERG V5 performed slightly better than 3B42 V7 for the detection of daily precipitation at the measuring station. The 3B42 V7 and IMERG V5 products showed a strong correlation with a high CC (0.78) and (0.81), a small RMSE error value (2.03 mm) and (1.68 mm), and a relatively balanced bias and R-bias (29.4%), (0.19) and (47.13%), (0, 31) respectively, the POD and CSI values are (1 vs. 1) and (0.89 vs. 0.88), the FAR and FBI obtained values are (0.1 vs. 0.12) and (1.1 vs. 1.13) respectively, noted that the results of the categorical measures are good.



(3) The performance evaluation results on the monthly scale show good performance of the SPPs. The 3B42 V7
show a better correlation with gauge rainfall than IMERG V5, with higher CC (0.80) and (0.75), low RMSE (19.90
mm vs. 22.09 mm), acceptable MAE (12.85) and (14.42), and relatively low Bias values (16.47% vs. 21.73%) and
(4.14 vs. 5.46) respectively. POD and CSI values are (1 vs. 1) and (0.89 vs. 0.88), FAR and FBI results are (0.1
vs. 0.12) and (1.1 vs. 1.13) respectively. However, 3B42 V7 performed better than IMERG V5 on the categorical
measures at a monthly scale.
(4) Regarding annual performance, the two satellite products followed the trend of rain gauge observations very
closely, 3B42 V7 slightly underestimating precipitation. 3B42 V7 and IMERG V5 present a strong correlation
with high CC values of (0.90) and (0.85), low RMSE error (42.84 mm) and (53.64mm), small MAE (34.72) and
(29.03), and relatively good R Bias and Bias (-2.99% vs. 4.07%) and (-10.61 and 14.44) respectively, for the R
Bias and Bias, the negative deviation of 3B42 V7 precipitation estimates were relatively balanced, while IMERG
V5 showed a positive deviation at the rain gauge. The values of the POD and the CSI are (1 vs. 1) and (1 vs. 1),
the FAR and FBI values are (0 vs. 0), and (1 vs. 1) respectively. These results exhibit perfect values. Indeed,
IMERG V5 showed better performance than 3B42 V7 in the statistical measures on an annual time scale.
(5) The hydrological simulations and calibration were performed according to two different scenarios; scenario 1
aims to run the model with observed rainfall and runoff data, scenario 2 used the SPPs with observed flows. The
obtained results are satisfactory for all simulations with different precipitations inputs. The Nash coefficients are
very good, ranging from 53.2% to 95.5% for the 3B42 V7, IMERG V5, and observed precipitation respectively.
The main point to remember is that both satellite precipitation products tend to underestimate precipitation during
wet seasons and overestimate them during dry seasons. The proposed method is an interesting approach to apply
for solving the problem of insufficient observed data in the Mediterranean regions.
Therefore, the results of this study are of great importance for analyzing the prospect's application of SPPs at
different time scale, this paper is one of the first papers developing a comparative approach of satellite rainfall
products to observed gauge data in North Africa, they could indeed serve researchers as a reference work both in
Morocco and neighbouring countries with similar climates and areas with irregular or sparse rain gauge networks.
**Acknowledgement:** The authors would like to thank Prof. Adam Milewski (Director of Water Resources &
Remote Sensing Laboratory (WRRS), Department of Geology, University of Georgia, United States), who
thoughtfully revised this manuscript.
This research has been supported by "PRIMA-S2-ALTOS-2018 Managing water resources within Mediterranean
agrosystems by Accounting for spatial structures and connectivity", and ERANETMED3-062 CHAAMS: global
Change: Assessment and Adaptation to Mediterranean region water Scarcity.

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
