# Peer review of "(IMERG V5) satellite precipitation products over a"

_Hydrology and Earth System Sciences, 2021_

## Author Comment (AC1)

**Response to Reviewer #1 Comments**

**Manuscript Number: HESS-2021-243**

**We are very grateful to the reviewer's for his deep and thorough review of our manuscript, we also thank him for the effort and time put into the review of the manuscript. We revised our present research in the light of their useful remarques; the comment has been carefully considered and responded. Hope our revision has improved to a level of their satisfaction, and our manuscript will be accepted in this new form.**

**Comment 1**: I found a mismatch in the GPM IMERG V5 satellite data. This version starts in 2014 until now. How did you work on the 2010-2014 period? Whereas the product only covers the period 2014-2017 of your data series.

**Response**: We appreciate your comment, GPM has up-graded its data algorithms to calibrate and incorporate data from the Tropical Rainfall Measurement Mission (TRMM) into its data record. NASA's Integrated Multi-satellitE Retrievals for GPM (IMERG) algorithm merges data from the TRMM and GPM missions, giving meteorologists and researchers access to a 20 years record of precipitation (from June 1, 2000 to present). Where previously data users had to work with two different types of precipitation numbers in a dataset (one for TRMM and one for GPM), IMERG allows users the ability to use TRMM and GPM data in an integrated manner for long-term studies.

The implemented strategy is to take all TRMM data and apply the IMERG algorithm to obtain a historical data series combine from the previously existing data. "Dr. Dalia Kirschbaum."

**Comment** 2: The authors did not justify the choice of precipitation products, why TRMM and GPM? Are these products never used in flood modeling in Morocco?

**Response**: Thank you for your specific comment; the TRMM and GPM satellite precipitation products are highly reputed for their quality of precipitation estimation, and have been widely used for comparison with observed data in Morocco (El-Bouhali et al., 2020) and around the world, as well as used for hydrological modeling in several regions with different climates (Yuan et al., 2019), (Furl et al., 2018), (Tong et al., 2014).However, these products were used for comparison and hydrological modeling for the first time in the Moroccan High Atlas in this study, due to the lack of precipitation measurement stations and the difficulty of access to the area due to the steep slope and high altitude of the mountains.

**Comment 3**: I do not see a detailed introduction that points the problem issue that the paper can solve.

**Response**: The reviewer brings up an important point that concerns the interest of this study to potential readers. This point is discussed in Line 87, Line 97: This study evaluated statistically and hydrologically the precipitation estimates of the GPM TRMM (3B42 V7) and (IMERG V5) satellites in relation to ground-based precipitation observations over the semi-arid Zat mountain catchment located in the Moroccan High Atlas, The aim of this research is to solve a major problem due to the irreparability of the precipitation measurement stations, which results in a

large data deficiency, thus leading to difficulties in working on such scientific aspects as flood forecasting and water management. The objectives are (1) to evaluate and statistically compare the performance of IMERG V5 and 3B42 V7 precipitation products at multiple temporal scales in the Zat basin, (2) to analyze the precipitation detection capability of the 3B42 V7 and IMERG V5 satellite sensors, and (3) to evaluate the ability of the PPSs to reproduce rainfall events and demonstrate their ability to provide meaningful information in hydrological modeling and flood forecasting. This manuscript is a valuable reference for monitoring and forecasting rainfall in mountainous regions characterized by a Mediterranean climate, as well as in basins where rainfall stations are scarce or poorly distributed.

**Comment** 4: The daily time step is not recommended for small basins (majority of Mediterranean basins) (<1000km²) where the response time is very small (Borga et al., 2008). How can this time step help in flood forecasting?

**Response**: As suggested, the daily time step is not adequate for small Mediterranean basin, taking into account the response time factor which is quite low. Moreover, we adopted a new time step (3H) more suitable for the flood forecasting in the study area.

**Comment** 5: The statistical evaluation of precipitation products is an essential step before their use. The threshold you used at the daily time scale is correct (0.5mm) but you cannot use it at the monthly and yearly time step. Because it will always be detected by satellite products. You have to use one threshold for each time step.

**Response**: As recommended, the threshold of (0.5mm) was used only at the hourly and daily time scale. No threshold was applied at the monthly and annual time scale.

**Comment** 6: Figure 5: How did you get the boxplot? You only have one coefficient correlation value for each time step.

**Response**: Indeed, we are absolutely agree with this comment, we have only one value of the coefficient of correlation for each time step. Although, the boxplot was elaborated by taking into account the coefficient correlation of the satellite precipitation data, it would not be a valid approach to apply because it does not make sense and was deleted.

**Comment** 7: I do not understand how you chose the event that represents a season? Does the magnitude of the floods change from season to season? Can you clarify more this point with references?

**Response**: This comment highlights an important logic of this study, the selection of the events was done based on the results of a paper currently in progress. The Zat watershed has a seasonal influence which is clearly reflected in the flow amplitude. On this area no previous research has been carried out to refer to this subject, the following article will be the first reference for future work on this region.

Indeed, as the time step has been changed from daily to hourly, the choice of the new selected episodes has been made taking into account the availability of measured data (rainfall and flow), due to the lack of measuring stations we have several gaps in recording data, that's why we have selected the episodes with a complete data series of rainfall and flow (Figure 2).

[Figure]

**Comment** 8: The calibration of flood events requires a validation step, otherwise, how can we justify that the product can be used in flood forecasting? You need to add a validation step.

**Response**: As suggested, this comment is pertinent and will allow us to clearly improve the quality of the manuscript. A validation step has been added in the chapter 4 to justify the use of satellite products for flood forecasting.

**Comment** 9: Figures 7, 8, 9, and 10 are just screenshots of the HEC HMS platform please provide clear figures.

**Response**: As you recommended, the quality of the figures has been improved in order to increase their visibility.

**Specific comments:**

**Introduction:**

**Comment** 1: Line 66: JXAX or JAXA?

**Response**: Line 69: We apologize for this typo. It has been corrected in the manuscript.

**Comment** 2: Line 80: is there no paper published in the same study area (High Atlas of Morocco)?

**Response**: Thank you for your remark; indeed, there is no previous study on this aspect in the High Atlas of Morocco. The following article will be the first reference for future work on this region.

**Comment** 3: Line 85: "Observations" or observation. In the observed data section, you showed only one rain gauge.

**Response**: Line 80: We apologize for this typing mistake. It has been corrected in the manuscript.

**Comment** 4: Line 88: "Analyze the precipitation detection ability of 3B42 V7 and IMERG V5 satellite sensors" Please keep the same order of your products as in line 87.

**Response**: Line 92: The remark has been corrected.

**Study Area and datasets:**

**Comment** 1: Line 95 and Figure 1: Please locate the Tensift basin and the highest mountain of

Toubkal. Is it in the same basin?

**Response**: As recommended, this comment was answered in the Line 113, Figure 1: The Tensift basin is a watershed located in the High Atlas mountain massif, of which the highest mountain is Toubkal. The Figure 1, has been modified to include the Tensift basin on the map of geographical location.

[Figure]

**Comment** 2: Line 97: ''is located'' instead of ''is found''

**Response**: Line 102: The remark has been corrected.

**Comment** 3: Line 100: ''Upstream" instead of "Downstream"

**Response**: Line 105: The remark has been corrected.

**Comment** 4: Line 104: Average Range? Please provide one average value of rainfall.

**Response**: Line 109: The remark has been corrected.

**Comment** 5: Line 106: Add a reference.

**Response**: The reference has been added

**Comment** 6: Line 113: ''These stations". You are working with one rain gauge please be clear in the data selection.

**Response**: Line 118: We apologize for this typing mistake. It has been corrected in the manuscript.

**Comment** 7: Line 119: You don't need to convert UTC to summertime (UTC+1) it is not the case for Morocco.

**Response**: Thank you for your specific comment, it has been taken into account and the conversion has been corrected.

**Comment** 8: Line 121: The GPM product started in March 2014. How did you evaluate the product between September 2010 and February 2014?

**Response**: We appreciate your comment, GPM has up-graded its data algorithms to calibrate and incorporate data from the Tropical Rainfall Measurement Mission (TRMM) into its data record. NASA's Integrated Multi-satellitE Retrievals for GPM (IMERG) algorithm merges data from the TRMM and GPM missions, giving meteorologists and researchers access to a 20 years record of precipitation (from June 1, 2000 to present). Where previously data users had to work with two different types of precipitation numbers in a dataset (one for TRMM and one for GPM), IMERG allows users the ability to use TRMM and GPM data in an integrated manner for long-term studies.

The implemented strategy is to take all TRMM data and apply the IMERG algorithm to obtain a historical data series combine from the previously existing data. "Dr. Dalia Kirschbaum."

**Comment** 9: Line 121: (Table 1) please keep the same order of products

**Response**: Table 1 has been deleted, but the comment has been taken into account, the order has been respected throughout the new version of the manuscript.

**Comment** 10: Line 122 and Line 131: Please keep the same order of products

**Response**: The order has been respected throughout the new version of the manuscript.

**Comment** 11: Line 126: Hou et al., 2014 reference is about the GPM product, not the TRMM

**Response**: Indeed, this reference and for the GPM, it has been removed.

**Comment** 12: Line 144: The website is for TRMM, not GPM

**Response**: Line 132, line 146: Please here is the website used to download the digital data of the two satellite products. (https://pmm.nasa.gov/data-access/downloads/trmm).

**Methodologies:**

**Comment** 1: Line 150: What do you mean by: "increasing the point precipitation data"?

**Response**: Line 152: The correct expression is to plot, we are sorry for this error of expression.

**Comment** 2: Line 151: The TRMM product has only one pixel over the basin. How it is representative to compare one pixel to 6 pixels for GPM. Also, you need to mention the number of pixels for each product over the study basin.

**Response**: Line 151, Line 155: This paragraph only includes an explanation of the comparison methods used in this type of approach. In our case, we used a direct comparison of the numerical data downloaded from the platform.

**Comment** 3: Line 152: The interpolation method can be only used for the GPM product, not for the Rain gauges

**Response**: Line 155: In our case, we used a direct comparison of the numerical data downloaded from the NASA platform with the observed precipitation.

**Comment** 4: Line 156: replace "reproduce" with "estimate"

**Response**: Line 156: The remark has been corrected.

**Comment** 5: Line 157: "the rainfall measurements stations". Please, be precise: you use only one gauge station.

**Response**: Line 157: The remark has been corrected.

**Comment** 6: Line 164: What observations?

**Response**: Line 164: We mean by ground observation, the gauge data measured in situ.

**Comment** 7: Line 168: Delete "Including"

**Response**: Line 168: The remark has been corrected.

**Comment** 8: Line 170: The Pearson Correlation Coefficient is not the same as the correlation coefficient in line 168?

**Response**: Line 168, Line 169: Indeed, the Correlation Coefficient is the same as Pearson Correlation Coefficient, it is a redundancy, and this paragraph has been deleted.

**Comment** 9: Table 2: Bias is in mm. You divided by the time length and the Ratio is not a unit.

**Response**: Line 170, Table 1: The remark has been corrected.

**Comment** 10: Line 199: how did you select the events that represent each season? Please clarify his point

**Response**: Due to the lack of measuring stations we have several gaps in recording data, that's why we have selected the episodes with a complete data series of rainfall and flow.

**Comment** 11: Figure 2: Delete the underlines in the input box

**Response**: Figure 2 has been deleted, and replaced by a text explanation to better explain the approach applied.

**Results:**

**Comment** 1: Figure 3A: The different plots of rainfall sources must be with a dotted line for one or two products to compare between them and to identify them. With the continued line the observed precipitation is hidden by the TRMM and GPM products.

**Response**: Figures 3 (a, b, c): have been deleted as they contain the same information as Table 2, to avoid repetition we have kept only the table 2 and 3.

**Table 2. Statistical metrics results of 3B42 V7 and IMERG V5 precipitation estimates at multiple time scales from 2012 to 2017.**

|  | TRMM | | | | GPM | | | |
|---|---|---|---|---|---|---|---|---|
|  | **3 Hours** | **Daily** | **Monthly** | **Yearly** | **3 Hours** | **Daily** | **Monthly** | **Yearly** |
| **CC** | 0,12 | 0,36 | 0,77 | 0,95 | 0,38 | 0,55 | 0,8 | 0,86 |
| **RMSE** | 1,5 | 1,03 | 2,18 | 16,75 | 1,48 | 1,24 | 3,01 | 22,41 |
| **Bais** | 0,21 | 0,26 | 0,33 | 0,22 | 0,38 | 1,22 | 1,46 | 1,49 |

**Table 3. Contingency statistical metrics results of 3B42 V7 and IMERG V5 precipitation estimates at multiple time scales from 2012 to 2017.**

|  | TRMM | | | | GPM | | | |
|---|---|---|---|---|---|---|---|---|
|  | **3 Hours** | **Daily** | **Monthly** | **Yearly** | **3 Hours** | **Daily** | **Monthly** | **Yearly** |
| **POD** | 0,13 | 0,35 | 0,92 | 1 | 0,36 | 0,76 | 1 | 1 |
| **FAR** | 0,67 | 0,65 | 0,07 | 0 | 0,79 | 0,79 | 0,08 | 0 |
| **CSI** | 0,1 | 0,21 | 0,86 | 1 | 0,15 | 0,18 | 0,91 | 1 |
| **FBI** | 0,4 | 1,01 | 1 | 1 | 1,82 | 3,82 | 1,09 | 1 |

**Comment** 2: How the GPM V5 product can be before 2014?

**Response**: GPM has up-graded its data algorithms to calibrate and incorporate data from the Tropical Rainfall Measurement Mission (TRMM) into its data record. NASA's Integrated Multi-satellitE Retrievals for GPM (IMERG) algorithm merges data from the TRMM and GPM missions, giving meteorologists and researchers access to a 20 years record of precipitation (from June 1, 2000 to present). Where previously data users had to work with two different types of precipitation numbers in a dataset (one for TRMM and one for GPM), IMERG allows users the ability to use TRMM and GPM data in an integrated manner for long-term studies.

The implemented strategy is to take all TRMM data and apply the IMERG algorithm to obtain a historical data series combine from the previously existing data. "Dr. Dalia Kirschbaum."

**Comment** 3: Figure 4A: How many zeros for each product compared to the observed precipitation. The main problem in the daily correlation is the number of zeros in the observed precipitation. Did you apply any kind of filter? Can you please, discuss a little bit the results by comparing the correlation results with other papers in Morocco?

**Response**: The number of 0's for each product compared to the observed precipitation are:

- Number of 0 for 3B42V7: Hourly time step 15346, daily time step 2217.
- Number of 0 for IMERGV5: Hourly time step 30080, daily time step 3608.
- Number of 0 for precipitation observe: hourly time step 14202, daily time step 1642.

No filter has been applied for the correlation at the analyzed time steps.

Based on the results of the article by (Bouhali et al., 2020), which is the only study in Morocco similar to the approach presented in this article. We find that at monthly time step the CC obtained is 0.7 while for our case the CC is on average 0.78, for the annual time step at the article of Bouhali et al., 2020 it has obtained an average CC of 0.77. While for our case we obtained an average annual CC of 0.90.

**Comment** 4: Figure 5: How did you get the boxplot and you only have one value per correlation?

**Response**: Indeed, we are absolutely agree with this comment, we have only one value of the coefficient of correlation for each time step. Although, the boxplot was elaborated by taking into account the coefficient correlation of the satellite precipitation data, it would not be a valid approach to apply because it does not make sense and was deleted.

**Comment** 5: Line 288: "High" instead of "low"

**Response**: Line 219: The remark has been corrected.

**Comment** 6: Figure 6: the high values of POD and low values of FAR at monthly and yearly time scale is related to the low threshold (0.5mm). Can you please discuss these results by comparing them to other papers in the same region?

**Response**: The high values of POD and FAR at monthly and yearly time scale is not due to the application of a threshold, because when we added the hourly time scale all the calculations were redone and no threshold was applied, although the values are still high.

**Comment** 7: Line 308: How can you identify that selected events are the most representative of the data series?

**Response**: Due to the lack of measuring stations we have several gaps in recording data, that's why we have selected the episodes with a complete data series of rainfall and flow Figure 2.

**Comment** 8: Table 6, 7, 8, and 9: Must be in the table list not integrated into the Figure 7, 8, 9, and 10.

**Response**: Table 4: We thank you for this comment, this detail has been taken into consideration.

**Comment** 9: Please consider adding the values of other parameters (Transfer Function and Recession).

**Response**: The table 4: has the following parameters, thank you for your comment.

**Comment** 10: The calibration of the parameters will automatically give good results please consider adding more events and consider adding the validation step.

**Response**: As you recommended Line 242, line 244: A validation step has been added to improve the quality of the manuscript. Thank you for your valuable comments and for the time you have given to correct this article.

**References:**

El Bouhali A., Lebaut S., Qadem A., Amyay M., Gille E., Cotonnec A., evaluation des produits TRMM et GPM à partir d'observations aux stations et de résultats d'un modèle de quantification spatiale des précipitations sur le moyen-atlas, Maroc. XXXIII ème Colloque de l'AIC Rennes 2020, changement climatique et térritoires, Jul 2020, Rennes, France. ffhal-02937327.

Furl C., Ghebreyesus D., O. Sharif H., Assessment of the Performance of Satellite-Based Precipitation Products for Flood Events across Diverse Spatial Scales Using GSSHA Modeling System. Geosciences. 2018, 8, 191; doi:10.3390/geosciences8060191

---

## Author Comment (AC2)

**Response to Reviewer #2 Comments**

**Manuscript Number: HESS-2021-243**

**We are very grateful to the reviewer's for his deep and thorough review of our manuscript, we also thank him for the effort and time put into the review of the manuscript. We revised our present research in the light of their useful remarques; the comment has been carefully considered and responded. Hope our revision has improved to a level of their satisfaction, and our manuscript will be accepted in this new form.**

**Comment** 1: it is not clear to me the final aim of the paper and I don't agree with the general conclusion drawn in the lines 428-433. For which applications a such analysis could be useful? Gauged-corrected satellite precipitation products, such as the ones used in the study, have a latency of several months and cannot be used for flood or precipitation forecasting as stated by the authors (see Lines 88-90 or 91-92, respectively). For the specific case study (small poorly gauged basin) it would be more meaningful to test the capability of near-real time satellite precipitation products in reproducing rainfall and streamflow time series at hourly time step.

**Response**: The reviewer brings up an important point that concerns the interest of this study to potential readers, indeed your comment has been taken into account, thus a time step of 3H has been added to the analysis and used for the modeling of precipitation events.

This comment highlights an important logic of this study, this point is discussed in Line 87, Line 97: The study evaluated statistically and hydrologically the precipitation estimates of the GPM TRMM (3B42 V7) and (IMERG V5) satellites in relation to ground-based precipitation observations over the semi-arid Zat mountain catchment located in the Moroccan High Atlas, The aim of this research is to solve a major problem due to the irreparability of the precipitation measurement stations, which results in a large data deficiency, thus leading to difficulties in working on such scientific aspects as flood forecasting and water management. The objectives are (1) to evaluate and statistically compare the performance of IMERG V5 and 3B42 V7 precipitation products at multiple temporal scales in the Zat basin, (2) to analyze the precipitation detection capability of the 3B42 V7 and IMERG V5 satellite sensors, and (3) to evaluate the ability of the PPSs to reproduce small rainfall events and demonstrate their ability to provide meaningful information in hydrological modeling and flood forecasting.

**Comment** 2: If I well understood from lines 153-155, the comparison in terms of rainfall was made by comparing the in situ data against the data extracted from the pixel covering the in situ station. If so, how are the authors considering the different spatial resolution of TRMM 3B42 and IMERG V5? Moreover, how the satellite precipitation data are extracted for the flood simulation?

**Response**: Thank you for your remark; Line 151, Line 155: only explains the comparison methods used in the bibliography of this kind of approach. In our case, we used a direct comparison of the numerical data downloaded from NASA website (https://pmm.nasa.gov/data-access/downloads/trmm) with the in-situ data.

**Comment** 3: It is not specified which IMERG product (Early, Late or Final run) is used in the analysis

**Response**: We appreciate your comment, in this study we used IMERGV5 Final run recommended for general use.

**Comment** 4: some Figures and Tables, e.g. Figure 3 and 4 or Figure 6 and Table 5, represent the same information. Please, remove one of them in the revised version of the manuscript.

**Response**: As you recommended; all figures have been removed, Tables 2 and 3 illustrate all information related to the statistical analysis.

**Table 2. Statistical metrics results of 3B42 V7 and IMERG V5 precipitation estimates at multiple time scales from 2012 to 2017.**

|  | TRMM | | | | GPM | | | |
|---|---|---|---|---|---|---|---|---|
|  | **3 Hours** | **Daily** | **Monthly** | **Yearly** | **3 Hours** | **Daily** | **Monthly** | **Yearly** |
| **CC** | 0,12 | 0,36 | 0,77 | 0,95 | 0,38 | 0,55 | 0,8 | 0,86 |
| **RMSE** | 1,5 | 1,03 | 2,18 | 16,75 | 1,48 | 1,24 | 3,01 | 22,41 |
| **Bais** | 0,21 | 0,26 | 0,33 | 0,22 | 0,38 | 1,22 | 1,46 | 1,49 |

**Table 3. Contingency statistical metrics results of 3B42 V7 and IMERG V5 precipitation estimates at multiple time scales from 2012 to 2017.**

|  | TRMM | | | | GPM | | | |
|---|---|---|---|---|---|---|---|---|
|  | **3 Hours** | **Daily** | **Monthly** | **Yearly** | **3 Hours** | **Daily** | **Monthly** | **Yearly** |
| **POD** | 0,13 | 0,35 | 0,92 | 1 | 0,36 | 0,76 | 1 | 1 |
| **FAR** | 0,67 | 0,65 | 0,07 | 0 | 0,79 | 0,79 | 0,08 | 0 |
| **CSI** | 0,1 | 0,21 | 0,86 | 1 | 0,15 | 0,18 | 0,91 | 1 |
| **FBI** | 0,4 | 1,01 | 1 | 1 | 1,82 | 3,82 | 1,09 | 1 |

**Comment** 5: it is not clear to me how the authors construct Figure 5. If I well understood, Figure 5 illustrates the results of Table 4. How the authors built the boxplots?

**Response**: Indeed, we are absolutely agree with this comment, we have only one value of the coefficient of correlation for each time step. Although, the boxplot was elaborated by taking into account the coefficient correlation of the satellite precipitation data, it would not be a valid approach to apply because it does not make sense and was deleted.

**Comment** 6: some indexes (e.g., Nash-Sutcliffe) are not defined in the text.

**Response**: We thank you for this comment, this detail has been taken into consideration.

**Comment** 7: Figures 7-9 should be improved as they are hard to read.

**Response**: As recommended, all figures have been changed and their quality has been increased.